**Data Availability Statement:** All relevant data are within the paper and its Supporting Information files.

# Prevalence of HER2 overexpression and amplification in cervical cancer: A systematic review and meta-analysis

**Boris Itkin**[1]*, **Agustin Garcia**[2], **Samanta Straminsky**[1], **Eduardo Daniel Adelchanow**[3], **Matias Pereyra**[4], **Gabriela Acosta Haab**[5], **Ariel Bardach**[6]

**1** Department of Oncology, Juan A Fernández Hospital, Buenos Aires, Argentina, **2** Department of Oncology, María Curie Hospital, Buenos Aires, Argentina, **3** Department of Oncology, Hospital Nacional Profesor Alejandro Posadas, Morón, Argentina, **4** Department of Pathology, Juan A Fernández Hospital, Buenos Aires, Argentina, **5** Department of Pathology, María Curie Hospital, Buenos Aires, Argentina, **6** Center for Research in Epidemiology and Public Health, Institute for Clinical Effectiveness and Health Policy (IECS)—National Scientific and Technical Research Council—Argentina, Buenos Aires, Argentina

* borisitkinl@gmail.com

## Abstract

The reported rates of HER2 positivity in cervical cancer (CC) range from 0% to 87%. The importance of HER2 as an actionable target in CC would depend on HER2 positivity prevalence. Our aim was to provide precise estimates of HER2 overexpression and amplification in CC, globally and by relevant subgroups. We conducted a PRISMA compliant meta-analytic systematic review. We searched Medline, EMBASE, Cochrane database, and grey literature for articles reporting the proportion of HER2 positivity in CC. Studies assessing HER2 status by immunohistochemistry or in situ hybridization in invasive disease were eligible. We performed descriptive analyses of all 65 included studies. Out of these, we selected 26 studies that used standardized American Society of Clinical Oncology / College of American Pathologists (ASCO/CAP) Guidelines compliant methodology. We conducted several meta-analyses of proportions to estimate the pooled prevalence of HER2 positivity and subgroup analyses using geographic region, histology, tumor stage, primary antibody brand, study size, and publication year as moderators. The estimated pooled prevalence of HER2 overexpression was 5.7% (CI 95%: 1.5% to 11.7%) $I^2$ = 87% in ASCO/CAP compliant studies and 27.0%, (CI 95%: 19.9% to 34.8%) $I^2$ = 96% in ASCO/CAP non-compliant ones, p < 0.001. The estimated pooled prevalence of HER2 amplification was 1.2% (CI 95%: 0.0% to 5.8%) $I^2$ = 0% and 24.9% (CI 95%: 12.6% to 39.6%) $I^2$ = 86%, respectively, p = 0.004. No other factor was significantly associated with HER2 positivity rates. Our results suggest that a small, but still meaningful proportion of CC is expected to be HER2-positive. High heterogeneity was the main limitation of the study. Variations in previously reported HER2 positivity rates are mainly related to methodological issues.

**Funding:** The authors received no specific funding for this work.

**Competing interests:** The authors have declared that no competing interests exist.

## Introduction

Cervical cancer (CC) is a significant global public health problem [1]. In 2020, more than 340 000 deaths were attributed to CC worldwide. It is the fourth leading cause of cancer-related mortality in women overall and the second among those under the age of 50 [2]. Human Papilloma Virus (HPV) is the main etiological factor of cervical carcinogenesis. It produces DNA damage, centrosome abnormalities, epigenetic changes, and DNA methylation [3] {Gupta, 2019 #391}. Despite the development of effective methods of primary (prophylactic anti-HPV vaccines) and secondary (screening for premalignant cervical lesions) prevention, it has been estimated that in low-income countries, where the proportion of patients with advanced disease is particularly high [4, 5], a substantial reduction of the CC burden may take several decades [1]. Currently, the standard treatments available for progressive, recurrent, and metastatic disease are limited, highlighting the need for new therapeutic options [6–8].

Enhanced signaling via the HER2 receptor plays a crucial role in cellular transformation, carcinogenesis, and maintenance of malignant phenotype and is considered an ideal target for antitumor treatments [9]. It may occur either due to an activating mutation, amplification of the HER2 gene or its amplification or otherwise, because of overexpression of the HER2 protein. *HER*2 gene somatic mutations in CC were proved to be a promising target for specific inhibitors in preclinical models and clinical studies [10–12]. They are beyond the scope of this work. On the other hand, several drugs targeting HER2 amplification and overexpression are available. HER2 targeting drugs have demonstrated clinical efficacy against HER2-positive breast, gastroesophageal, and serous endometrial cancers, being the current therapeutic standards for these conditions and becoming investigational treatments in a continuously expanding set of solid tumors [13–17]. However, to our knowledge, no clinical trial assessing HER2 amplification/overexpression targeting agents in CC has been published to date, and the evidence of their successful use in this disease is limited to case reports [18]. The suitability of HER2 as an actionable target for clinical studies in CC would depend on HER2 positivity prevalence in this disease. Further knowledge about the frequency of HER2 overexpression/amplification in CC is required.

Standards for the accurate evaluation of HER2 status and the proper definition of HER2 positivity has evolved over time and vary across tumor types. In early clinical trials of trastuzumab for breast cancer, HER2 status was determined by immunohistochemistry (IHC), and a 3 + score corresponded to "more than moderate," i.e., strong, staining of the "entire tumor-cell membrane. . . in more than 10 percent of tumor cells" [19]. HER2 gene amplification, defined as a HER2 gene to chromosome 17 (HER2 /CEP17) ratio of at least 2.0 by Fluorescent In Situ Hybridization (FISH), was also a part of the positivity definition endorsed by the US Food and Drug Administration [20]. In 2007, the American Society of Clinical Oncology and the College of American Pathologists (ASCO/CAP) released guidelines that defined as positive those cases with >30% of invasive tumor cells with a uniform intense membrane staining by IHC or HER2 /CEP17 of >2.2 for dual probes or >6 HER2 gene copies for single probes by FISH. The updated 2013 ASCO/CAP recommendations returned the IHC positivity threshold to a 10% cut-off point and changed In Situ Hybridization (ISH) grading criteria [21, 22]. It was estimated that these changes in the scoring system would lead to a reclassification of 7.7% of patients scored by FISH and 3.7% of patients graded by IHC [20, 21]. The 2018 focused update of the ASCO/CAP Guidelines addressed some rare clinical situations and allowed primary ISH [23]. A slightly different IHC 3+ definition is used for gastric cancer, as apical membranous staining is not required [24].

Since 1990, many studies assessed HER2 receptor status in CC. The reported proportion of HER2-positive tumors ranged widely between 0 and 87% [25–30]. To our knowledge, factors

accounting for such a disparity have not been formally assessed, and hitherto no systematic review on the topic has been published. As guidelines for HER2 assessment in cervical cancer have not been developed, some studies utilized one of the existing ASCO/CAP guidelines for breast and gastroesophageal cancers as a reference point.

Hence, our objectives were to estimate the pooled prevalence of the HER2 amplification and overexpression in patients with invasive CC, globally and according to the HER2 scoring system and positivity definition using meta-analysis to increase the statistical precision. Our aim was to determine whether HER2 status varies across different histologic subtypes and clinical stages, identifying the subgroup of patients with a higher proportion of HER2 positivity and exploring associations between HER2 positivity and demographic, clinicopathological, and assay-related variables. Our research questions were:

a. How common is HER2 overexpression/amplification in patients with invasive CC?

b. Is the proportion of HER2-positive CC significantly different in studies that used ASCO/CAP compliant methodology and those that used other methods? and

c. Is there a relationship between HER2 pooled prevalence and relevant subgroups, such as histologic subtype, World Health Organization (WHO) geographic region, tumor stage, primary antibody brand, and year of publication?

## Materials and methods

The present study protocol was registered in the University of York's PROSPERO International Prospective Systematic Reviews Database under ID: CRD42018096078 on June 11, 2018 [31]. Eligible studies were those with any epidemiologic design and year of publication, published in any language, reporting overexpression and/or amplification of HER2 in CC by IHC or ISH in patients with histopathologically confirmed invasive CC, regardless of age, ethnicity, or geographic region. We allowed the inclusion of microinvasive carcinomas and did not impose restrictions regarding the treatment used before tissue collection. We excluded narrative reviews, comments, letters, editorials, case reports, and studies assessing *in vitro* and *in vivo* models. Also, we excluded studies with less than ten patients, those reporting HER2 status in preinvasive epithelial cervical lesions, lymphoid neoplasms, and melanoma of the uterine cervix, studies missing relevant data, and those publications whose access to full text was unavailable. If a study used IHC for HER2 protein overexpression followed by a non-ISH method for HER2 amplification assessment, only data on IHC were included in the review.

### Search strategies for the identification of studies and data sources

We conducted a search in Medline, EMBASE, LILACS, Cochrane and Google Scholar search engines with no language and date restrictions (up to December 22, 2020) using the following syntax: (Receptor, ErbB-2[Mesh] OR ErbB-2[tiab] OR CD340[tiab] OR Proto-Oncogene Protein*[tiab] OR HER-2[tiab] OR Neu Receptor*[tiab]) AND (Uterine Cervical Neoplasms [Mesh] OR Cervical Neoplas*[tiab] OR Cervical Cancer[tiab] OR Cervical Tumor*[tiab] OR Cervical Carcinom*[tiab] OR Cervix Neoplas*[tiab] OR Cervix Cancer[tiab] OR Cervix Tumor*[tiab] or Cervix Carcinom*[tiab] OR Cervical Adenocarcinom*[tiab] OR Cervix Adenocarcinom*[tiab] OR Cervical Intraepithelial Neoplasia[Mesh] OR Cervical Intraepithe- lial[tiab] OR Cervix Intraepithelial[tiab]). We translated the syntax into the different databases accordingly. We searched lists of references from relevant primary studies, reviews, and key journals for additional studies. Likewise, we explored books and grey literature, master/doc- toral theses, and meeting procedures. Automation tools were not used (See S1 File for details).

## Data management

We used Cochrane's web-based systematic review data management Covidence software to handle the initial phases of this review [32]. If duplication of a study report was the concern, we kept the larger one, with better methodological quality, and/or longer follow-up, as agreed by the entire team of investigators.

## Study selection and data collection

After the initial screening of titles and abstracts, a second round of screening by full text was performed according to the eligibility criteria. Selected papers were qualitatively described. We considered only studies that used a methodology compliant with ASCO/CAP guidelines for the quantitative synthesis. Each step of the study selection and data extraction process was carried out by at least two independent reviewers (BI, SS, EA, and AG). Disagreements, if detected, were referred to a third author or solved by consensus of the entire team. If additional information to resolve questions about eligibility was required, authors of articles were contacted by email. Reasons for exclusion of all the ineligible studies were recorded. The study flowchart is shown in Fig 1.

The proportions of HER2-positive tumors by IHC and ISH were the co-primary outcomes. We extracted information on a pre-piloted spreadsheet. This comprised geographic location, study design, patients' age, tumor stage, histology, sample, and assay characteristics, including

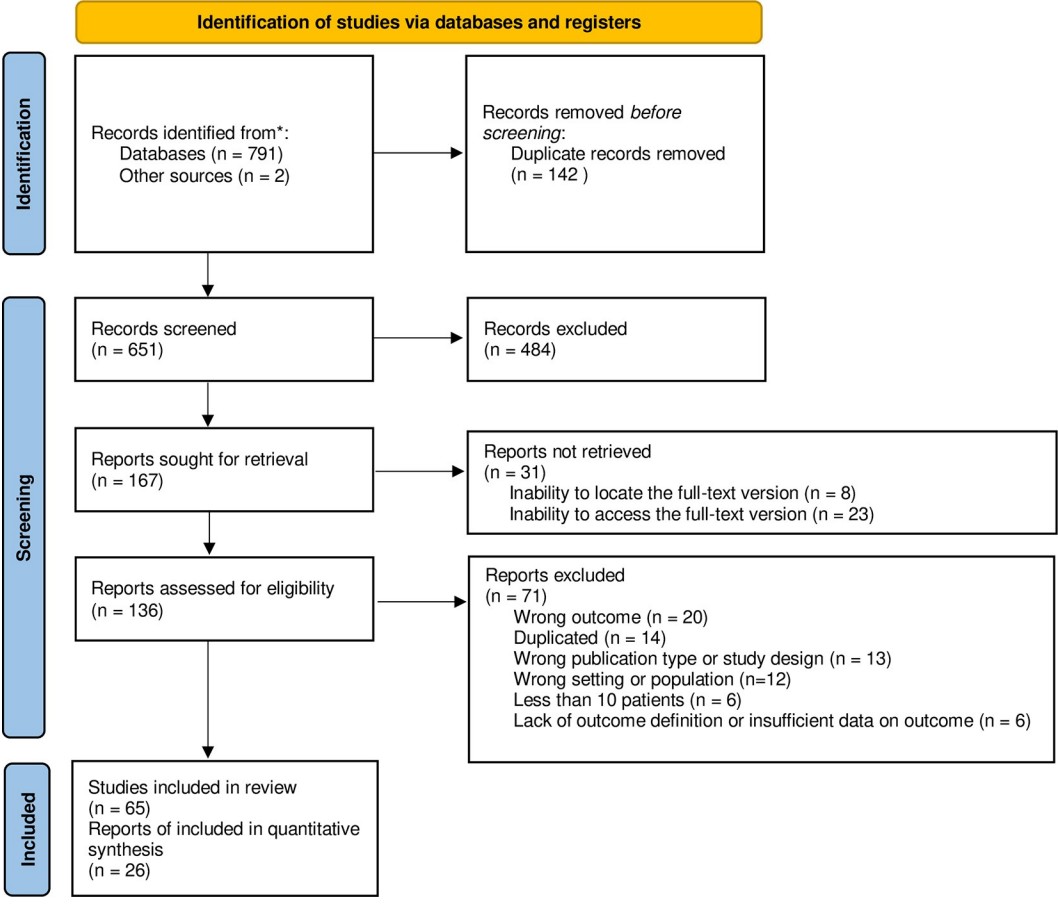

**Fig 1. PRISMA diagram of the study selection process.**

brands and clones of primary antibodies and probes, as well as criteria used by authors of included studies for the definition of HER2 positivity. The full-length list of extracted variables is available in the S2 File.

## Risk of bias assessment

We used the checklists of the National Institutes of Health Study Quality Assessment Tools for observational studies [33]. The methodology used for determining HER2 positivity was classified as ASCO/CAP compliant if the scoring system and positivity definition used in the study matched those made explicit in any ASCO/CAP guidelines for HER2 testing (2007, 2013, or 2018) for either breast or gastric cancer regardless of the year of study publication [22–24].

If a study had an ASCO/CAP compatible scoring system, but a different positivity definition (for example, both 2+ and 3+ were considered positive) and provided the data on the proportion of 3+ positive cases separately, it was also classified as ASCO/CAP compliant. Only the number of 3+ positive cases was used to calculate the proportion of HER2-positive tumors in such situations.

We hypothesized that the departure from ASCO/CAP standards might introduce bias, so when assessing the domain "outcome measurements", ASCO/CAP compliant studies were classified as low risk of bias and vice versa. While studies reporting on the prevalence of HER2 overexpression/amplification using any definition and cut-off values for HER2 positivity were eligible for qualitative synthesis, we restricted the quantitative synthesis to those ASCO/CAP-compliant ones. The same rule was not applied in the case of the domain "outcome assessment by two independent pathologists" since there was no statistically significant difference in the estimated pooled HER2 positivity prevalence between studies classified at low and high risk of bias in this domain, p = 0.81. We graphed funnel plots (S3 File) and reported Egger's test for publication bias appraisal. Adenocarcinomas, adenosquamous, glassy cell, clear cell, neuroendocrine and adenoid cystic carcinomas were grouped as non-squamous.

## Statistical analyses

We transformed proportions using the variance-stabilizing double arcsine square root Freeman-Tukey procedure. The between-study variance was computed by the Der Simonian and Laird method. For moderator analyses, we chose a mixed-effects approach, assuming a common between-study variance component [34]. A meta-regression technique was used for continuous moderators to describe the associations with the outcome variable HER2 positivity. Higgin's $I^2$ was used to assess heterogeneity alongside a visual inspection of the Forest plot and R-squared values for the proportion of between-study variance that each moderator could explain. To identify outliers and influential studies, we screened for studentized residuals absolute z-values close to or larger than 2. Leave-one-out analyses and Baujat plots were performed [34]. Statistical analyses were conducted with *meta* and *metafor* packages under R 3.6.3 or a later version [35]. Subgroup analyses were pre-specified in the study protocol for variables ASCO/CAP compliance and histology and exploratory for other moderators. All tests were conducted at a 0.05 alpha level.

## Results

### Study selection

After removing duplicates, we screened 651 documents, 649 from electronic databases, and two from grey literature (Fig 1) [36, 37]. Out of them, 136 were assessed for eligibility in full-

text in English, German, Chinese, Russian, and Turkish with the aid of the Google Translator tool. Finally, 65 were included in the review, as described. We tried to access full-text versions through our institutional libraries, Google Scholar, Research Gate, and by contacting the corresponding authors by email. Thirty-one reports could not be retrieved, eight because of the inability to locate the article, and 23 due to the unavailability of its full-text version or inability to pay for access owing to the lack of funding.

## Study characteristics

The included studies were published between 1990 and 2020, 64 of them in English and one in Russian [38]. Twenty-six studies were case series, 26 were retrospective cohorts, six were cross-sectional studies, three were case-control studies, three were prospective cohorts, and one study was a mixed cohort (see S4 File). As for their region of origin, 22 studies were from Europe, 18 were from the Asia Pacific Region, 14 were from the Americas, ten were from South-Eastern Asia, and one study was from the Eastern Mediterranean WHO Region (S5 File).

## Outcomes, assays, and measurements

The 65 studies included in the review (N = 5 237) were classified into three overlapping categories (Fig 2 and Table 1). The first subset comprised 62 studies that reported the prevalence of HER2 overexpression by IHC (N = 5 076). Fifty-four of them (N = 4 399) reported IHC only. In the remaining eight studies (N = 677), ISH was also performed in addition to IHC (N = 406).

The second subset included ten studies that reported the rates of HER2 amplification by ISH (N = 489) [26, 39, 42, 47, 52, 56, 67, 69, 82, 85]. In eight of them the HER2 overexpression was also determined by IHC and two studies (N = 83) used ISH as the only method [82, 85].

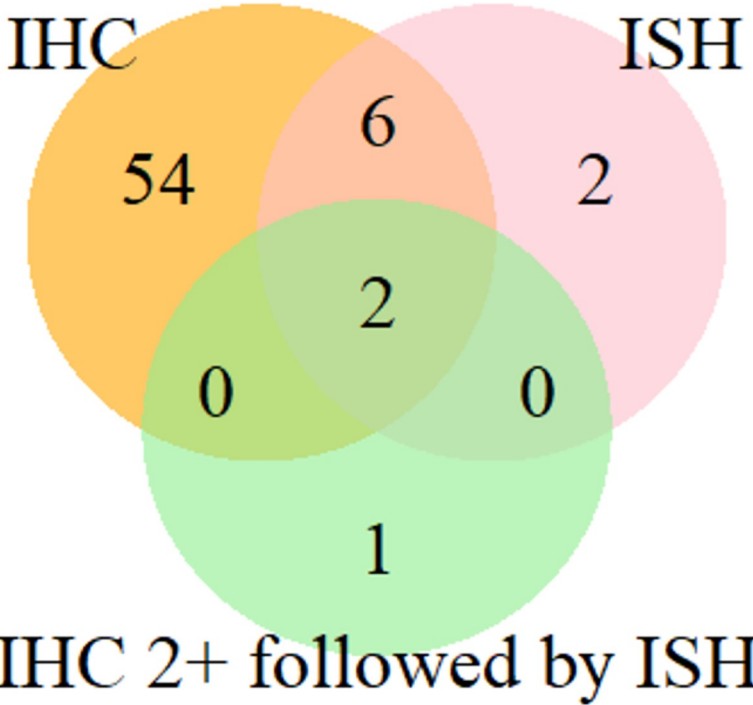

**Fig 2. Subgroups of studies according to the methods used for HER2 positivity determination.** Abbreviations: IHC = Immunohistochemistry, ISH = In Situ Hybridization.

Out of eight studies that reported results of both IHC and ISH, six studies (N = 394) [26, 39, 52, 56, 67, 69] performed ISH in patients not previously selected by IHC, and two studies (N = 170) [42, 47] only in those patients with equivocal results (2+) of IHC testing. In the third subset, there was only one study (N = 78) [62]. It reported the HER2 positivity prevalence by IHC tiebreaking the equivocal cases (2+) by ISH, but positivity rates by each method were not provided separately (Fig 2). No study performed IHC in patients pre-selected by ISH. Out of 11 studies which assessed HER2 positivity using ISH, nine performed FISH, one study chromogenic ISH (CISH), and the other one dual ISH (DISH) technique (S6 File).

Out of 62 studies reporting on HER2 positivity prevalence by IHC, 26 (41.9%) used the positivity definition and grading criteria consistent with ASCO/CAP 2007, 2013, or 2018 guidelines and have been classified as ASCO/CAP compliant, while the remaining 36 (58,1%) studies were classified as ASCO/CAP non-compliant (Fig 3 and Table 1). Among ASCO/CAP compliant studies, five used the 30% positivity cut-off point (ASCO/CAP 2007) [25, 38, 40, 48, 52], and 19 studies the 10% cut-off point (ASCO/CAP 2013 or 2018) [30, 36, 37, 39, 41, 42, 44, 45, 47, 49, 51, 55, 56, 58, 60, 65, 67, 70–72]. Nine out of 11 studies that used ISH followed ASCO/CAP compliant or slightly more stringent positivity criteria (S6 File).

## Subjects

The median age of subjects was 49.0 years old, (interquartile range [IQR] 45.0–51.0). Data on the tumor stage was available from 56 studies. Five studies included women with stage I only, 11 studies with stages I and II, two studies with stage III only. The remaining studies had a mixture of several stages. Median HPV positivity rate was 76.5%, (IQR: 57.3% to 90.3%). For more detailed information on patients' clinical features, see S7 File.

## Tumors and samples

In 14 studies (28.6%), tissue for the analyses was obtained by biopsy, in 25 (51.0%) by surgery, and the remaining ten studies (20.4%) used samples from either biopsy or surgery combined in varied proportions. The primary tumor was the exclusive sampling site in 41 (91.1%) studies. In the other 4 (8.9%) studies, in addition to the primary tumor, samples from nodes or distant or local recurrences were included. No study provided sampling-to-fixation and fixation-to-assay times. Data on histologic subtypes were available from 63 (96.9%) studies. Eighteen studies (28.6%) included patients with squamous carcinoma, 18 (28.6%) non-squamous histology, and 27 (42.9%) studies with both.

## Risk of bias assessment

The risk of bias assessment in studies is summarized in S8 File. In the tool utilized, the two domains of the potential bias most frequently involved were "outcome measurements" (60%) and "outcome assessment by two independent pathologists" (75%). Funnel plots for publication bias assessment are shown in S3 File. No significant asymmetry has been detected neither in the entire set of included studies (IHC, p = 0.769; ISH p = 0.543) nor in the subset of ASCO/CAP compliant ones, p = 0.936 suggesting the absence of substantial bias.

## Outcomes

**Immunohistochemistry.** Overall, the estimated pooled prevalence of HER2 overexpression was 17.0% (95% confidence interval [95%CI]: 11.7% to 23.0%), $I^2$ = 96% (S9 File). If only studies that used an ASCO/CAP compliant IHC method (26 studies, N = 2135) were included, under the random-effects model the estimated pooled prevalence of HER2 overexpression was

Table 1. Main characteristics of the included studies.

| Author | Year of publication | Country | Histology | Method used | ASCO/CAP compliance | N Analyzed IHC | % 3 + IHC | N Analyzed by ISH | % Positive by ISH |
|---|---|---|---|---|---|---|---|---|---|
| Shi [39] | 2020 | China | Ns | IHC, FISH | Yes | 209 | 18.2 | 209 | 6.2 |
| Varshney [40] | 2020 | India | Sq, Ns | IHC | Yes | 38 | 21.1 | 0 | |
| Wong [41] | 2020 | China | Ns | IHC | Yes | 14 | 21.4 | 0 | |
| Nakamura [42] | 2019 | Japan | Ns | IHC, DISH | Yes | 13 | 0 | 4 | 25.0** |
| Rahmani [43] | 2018 | Sudan | Sq, Ns | IHC | No | 65 | 43.1 | 0 | |
| Kumari Mitra [36] | 2018 | India | Sq, Ns | IHC | Yes | 30 | 10 | 0 | |
| Bajpai [44] | 2017 | India | Sq, Ns | IHC | Yes | 43 | 4.7 | 0 | |
| Halle [45] | 2017 | Norway | Sq, Ns | IHC | Yes | 292 | 20.8 | 0 | |
| Martinho [46] | 2017 | Brazil | Sq, Ns | IHC | No | 170 | 53.5 | 0 | |
| Ueda [29] | 2017 | Japan | Ns | IHC | No | 43 | 20.9 | 0 | |
| Xiang [47] | 2017 | China | NA | IHC, FISH | Yes | 157 | 2.5 | 8 | 25.0** |
| Carleton [48] | 2016 | UK | Ns | IHC | Yes | 26 | 3.8 | 0 | |
| Sarwade [37] | 2016 | India | Sq, Ns | IHC | Yes | 41 | 7.3 | 0 | |
| Nimisha Sharma [49] | 2016 | India | Sq, Ns | IHC | Yes | 25 | 4.0 | 0 | |
| Fukazawa [50] | 2014 | Brazil | Sq | IHC | No | 179 | 16.2 | 0 | |
| Nishio [51] | 2014 | Japan | Sq, Ns | IHC | Yes | 204 | 4.9 | 0 | |
| Vosmik [30] | 2014 | Czech Rep | Sq | IHC | Yes | 70 | 0 | 0 | |
| Barbu [25] | 2013 | Romania | Ns | IHC | Yes | 13 | 23.1 | 0 | |
| Conesa-Zamora [26] | 2013 | Spain | Sq | IHC, FISH | Yes | 32 | 3.1 | 32 | 0 |
| Khalimbekova [38] | 2013 | Russia | Clear cell | IHC | Yes | 14 | 0 | 0 | |
| Ueno [52] | 2013 | Japan | Clear cell | IHC, FISH | Yes | 13 | 23.1 | 8 | 12.5 |
| Sukpan [53] | 2011 | Thailand | NE | IHC | No | 100 | 2.0 | 0 | |
| Perez -Regadera [54] | 2010 | Spain | Sq, Ns | IHC | No | 136 | 23.5 | 0 | |
| Gupta [55] | 2009 | India | Sq, Ns | IHC | Yes | 65 | 27.6 | 0 | |
| Lesnikova [56] | 2009 | Denmark | Sq, Ns | IHC, CISH | Yes | 136 | 0.7 | 136 | 3.7 |
| Yamashita [57] | 2009 | Japan | Sq | IHC | No | 57 | 24 | 0 | |
| Shen [58] | 2008 | China | Sq | IHC | Yes | 53 | 0 | 0 | |
| Carreras [59] | 2007 | Spain | Sq | IHC | No | 10 | 50 | 0 | |
| Panek [60] | 2007 | Poland | Sq, Ns | IHC | Yes | 298 | 7.8 | 0 | |
| Protrka [61] | 2007 | Serbia | Sq | IHC | No | 13 | 46.2 | 0 | |
| Fuchs [62] | 2007 | Germany | Sq | IHC, FISH | No | 78 | ND | ND | 21.8* |
| Califano [63] | 2006 | Italy | Sq, Ns | IHC | No | 65 | 0 | 0 | |
| Kuroda [27] | 2006 | Japan | Glassy cell | IHC | No | 11 | 45.4 | 0 | |
| Ravazoula [64] | 2006 | Greece | Sq | IHC | No | 42 | 19.0 | 0 | |
| Kim [65] | 2005 | Korea | Sq, Ns | IHC | Yes | 258 | 0.4 | 0 | |
| Tangjitgamol [66] | 2005 | USA | NE | IHC | No | 24 | 0 | 0 | |
| Chavez -Blanco [67] | 2004 | Mexico | Sq, Ns | IHC, FISH | Yes | 35 | 2.9 | 4 | 0 |
| Graflund [68] | 2004 | Sweden | Sq, Ns | IHC | No | 172 | 5.2 | 0 | |
| Rosty [69] | 2004 | France | Sq, Ns | IHC, FISH | No | 82 | 2.4 | 5 | 0 |
| Bellone [70] | 2003 | USA | ND | IHC | Yes | 10 | 20.0 | 0 | |
| Dellas [71] | 2003 | Switzerland | Ns | IHC | Yes | 22 | 0 | 0 | |
| Heller [72] | 2003 | USA | Sq | IHC | Yes | 24 | 0 | 0 | |
| Niibe [73] | 2003 | Japan | Sq | IHC | No | 21 | 42.8 | 0 | |

(Continued)

**Table 1.** (Continued)

| Author | Year of publication | Country | Histology | Method used | ASCO/CAP compliance | N Analyzed IHC | % 3 + IHC | N Analyzed by ISH | % Positive by ISH |
|--------|--------------------|---------|-----------|-------------|--------------------|----------------|-----------|-------------------|-------------------|
| Kedzia [74] | 2002 | Poland | Sq | IHC | No | 47 | 4.3 | 0 | |
| Lee [75] | 2002 | Korea | Ns | IHC | No | 37 | 29.7 | 0 | |
| Bhadauria [76] | 2001 | India | Sq | IHC | No | 50 | 26.0 | 0 | |
| Leung [28] | 2001 | China | Ns | IHC | No | 78 | 87.2 | 0 | |
| Ngan [77] | 2001 | China | Sq | IHC | No | 101 | 19.8** | 0 | |
| Straughn [78] | 2001 | USA | NE | IHC | No | 16 | 0 | 0 | |
| Chang [79] | 1999 | China | Sq | IHC | No | 56 | 46.4 | 0 | |
| Kersemaekers [80] | 1999 | Netherlands | Sq, Ns | IHC | No | 132 | 9.1 | 0 | |
| Laksmi [81] | 1999 | India | Sq | IHC | No | 166 | 34.9 | 0 | |
| Mark [82] | 1999 | USA | Sq, Ns | FISH | No | 0 | | 23 | 8.7 |
| Nevin [83] | 1999 | UK | Sq, Ns | IHC | No | 126 | 38.1 | 0 | |
| Nishioka [84] | 1999 | UK | Sq, Ns | IHC | No | 107 | 32.7 | 0 | |
| Sharma [85] | 1999 | India | Sq | FISH | No | 0 | | 60 | 36.6 |
| Mandai [86] | 1997 | Japan | Sq, Ns | IHC | No | 88 | 38.6 | 0 | |
| Ndubisi [87] | 1997 | USA | Sq, Ns | IHC | No | 150 | 22.7 | 0 | |
| Kristensen [88] | 1996 | Norway | Glassy cell | IHC | No | 132 | 12.1 | 0 | |
| Nakano [89] | 1996 | Japan | Sq | IHC | No | 52 | 46.2 | 0 | |
| Costa [90] | 1995 | USA | Ns | IHC | No | 82 | 39.0 | 0 | |
| Kihana [91] | 1994 | Japan | Ns | IHC | No | 44 | 25.0 | 11 | |
| Oka [92] | 1994 | Japan | Sq, Ns | IHC | No | 192 | 19.3 | 0 | |
| Hale [93] | 1992 | UK | Sq, Ns | IHC | No | 62 | 38.7 | 0 | |
| Berchuck [94] | 1990 | USA | Sq, Ns | IHC | No | 33 | 9.1 | 0 | |

*Among all included patients.

**Among cases 2+ by immunohistochemistry.

Abbreviations: USA = United States of America, Czech Rep = Czech Republic, UK = The United Kingdom of Great Britain and Northern Ireland, Sq = squamous, Ns = non-squamous, NE = neuroendocrine, IHC = immunohistochemistry, FISH = fluorescence in situ hybridization, CISH = chromogenic in situ hybridization, DISH = dual in situ hybridization.

5.7%, (CI 95%: 1.5% to 11.7%), $I^2$ = 87% (Fig 3) [25, 26, 30, 36–42, 44, 45, 47–49, 51, 52, 55, 56, 58, 60, 65, 67, 70–72]. In the subset of studies considered ASCO/CAP non-compliant (36 studies, N = 2 941), the estimated pooled prevalence of HER2 overexpression was 27.0%, (IC 95%: 19.8% to 34.8%), $I^2$ = 96% (Fig 3).

In the mixed-effects model, the difference between these ASCO/CAP compliant and non-compliant subgroups was statistically significant (p < 0.001). The amount of heterogeneity accounted for (R2) has been estimated at 26.8%. Thus, for further analyses, we only considered the subgroup of ASCO/CAP compliant studies. As there was no statistically significant difference in the pooled prevalence of HER2 overexpression between subsets of studies that used ASCO/CAP 2007 and 2013/2018 cut-of points, p = 0.11 (not shown), we pooled them together for subsequent analyses. As significant heterogeneity in the ASCO/CAP compliant subgroup persisted, we conducted influence and moderator analyses. Although two studies have been identified as outliers, they were not considered influential cases since their removal neither significantly shifted the summary proportion nor markedly reduced the heterogeneity (S10 File) [45, 55].

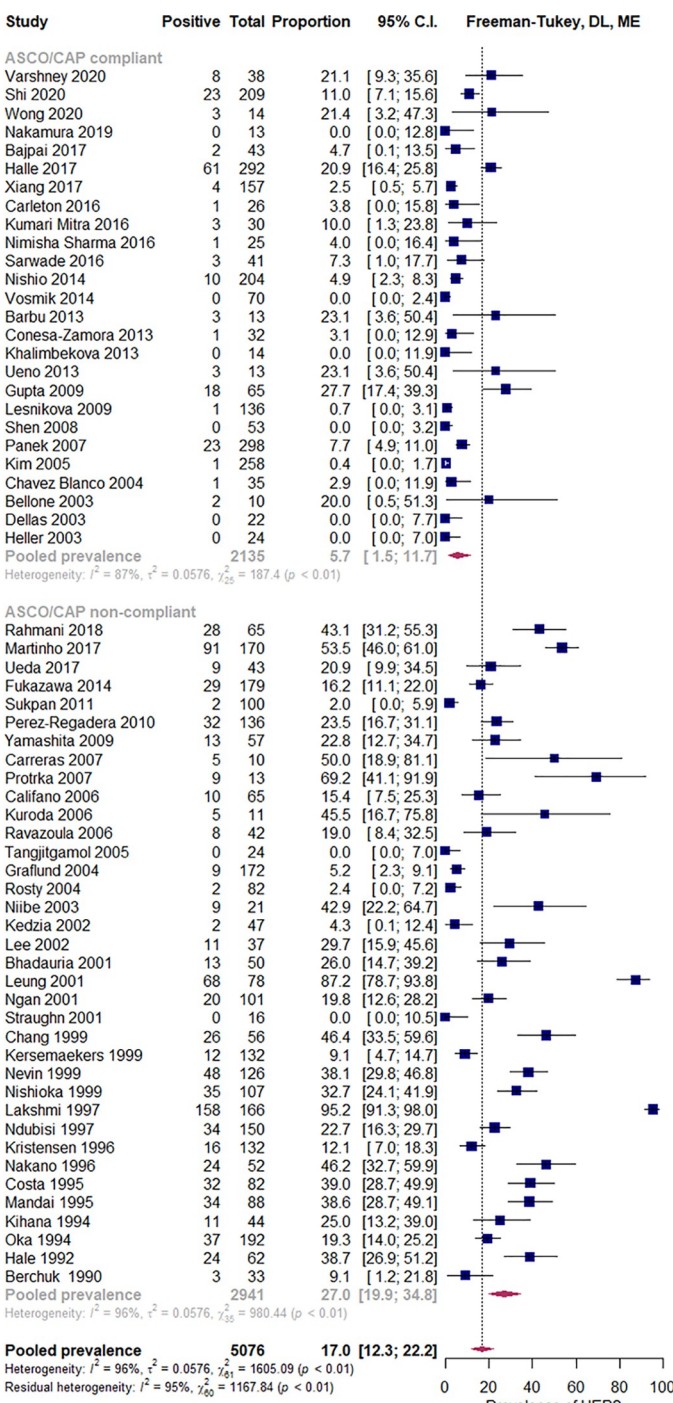

**Fig 3. Prevalence of HER2 overexpression according to ASCO/CAP-compliant guidelines.** Abbreviations: ASCO = American Society of Clinical Oncology, CAP = College of American Pathologists, DL = Der Simonian and Laird, ME = mixed effects.

**Moderator analyses.** In the squamous histology subgroup (12 studies, N = 1018), the estimated pooled prevalence of HER2 overexpression was 4.1% (CI 95%: 0.6% to 9.8%), $I^2$ = 91%, while in the non-squamous carcinoma studies (15 studies, N = 467) it was 10.3% (CI 95%: 3.6

to 19.2%), $I^2$ = 73%. If all the studies were considered, p = 0.054, R2 = 11%, (Fig 4), or if the analysis was restricted to only those studies which included both histologic subtypes, p = 0.12, R2 = 5.6% (Fig 5), there were no statistically significant differences. No statistically significant relationship was observed between pooled HER2 overexpression rate and predictor variables, i.e. geographic region (p = 0.40), primary antibody brand (p = 0.051), year of study publication (p = 0.067), study size (p = 0.871), and the proportion of the HPV positive tumors (p = 0.842). See Figs 6–8.

On the horizontal axis: year of study publication. On the vertical axis: double arcsine transformed proportion of HER2 overexpressing tumors in each study. The regression line is depicted in red. The size of each blue circle is proportional to the number of patients in the corresponding study.

**In situ hybridization.** In the subset of ASCO/CAP-compliant studies, the estimated pooled prevalence of HER2 amplification was 1.2% (CI 95% 0.0% to 5.8%) $I^2$ = 0% [26, 38, 39, 50, 52, 67, 69]. compared to 24.9% (IC 95% 12.6% to 39.6%), $I^2$ = 86% among the ASCO/CAP non-compliant ones [82, 85]. The difference was statistically significant, p = 0.004 (Fig 8). Two studies reported ISH positivity rates among HER2 2+ tumors [42, 47]. Two out of eight (25%) and one out of four (25%) patients, respectively, were positive by ISH (Fig 9).

**Special histologic subtypes and microinvasive carcinoma.** Varying degrees of HER2 overexpression have been observed in most histologic subtypes, except for mesonephric and

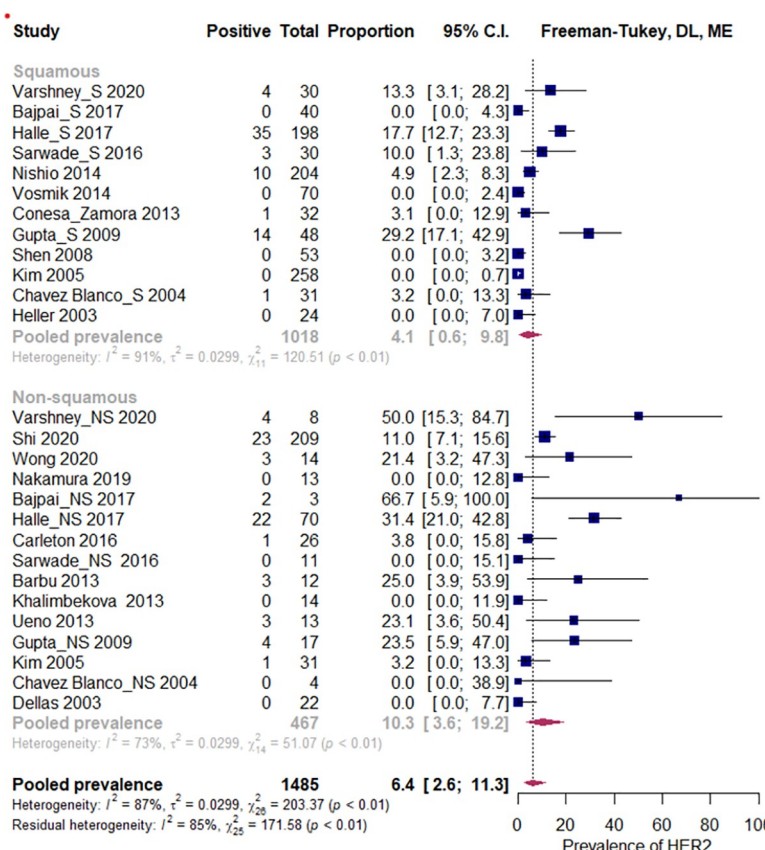

**Fig 4. Subgroup analysis by histologic subtype.** All studies. Abbreviations: DL = Der Simonian and Laird, ME = mixed effects.

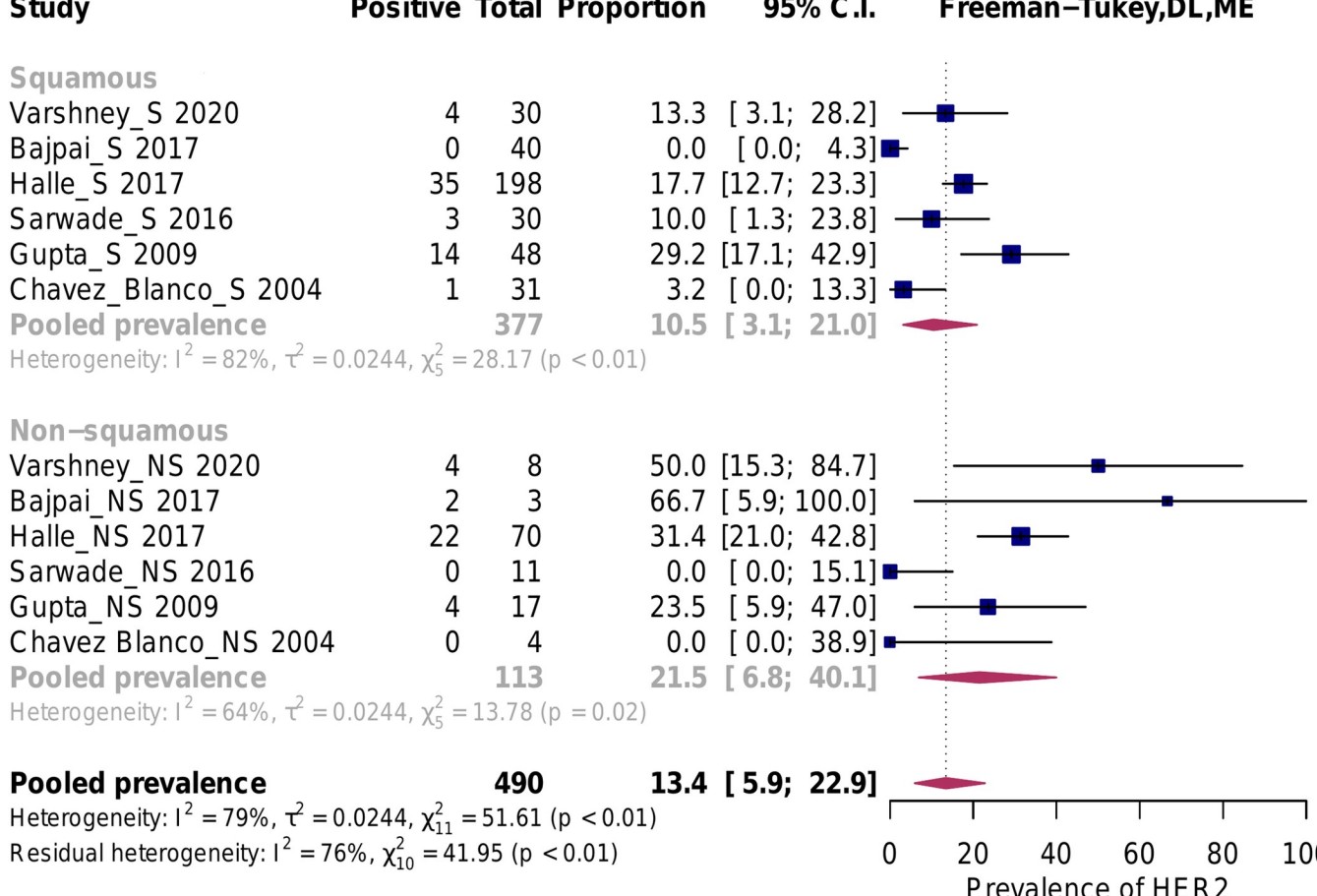

**Fig 5. Studies that included both squamous and non-squamous histology.** Abbreviations: DL = Der Simonian and Laird, ME = mixed effects.

perhaps neuroendocrine carcinomas (Fig 10 and S11 File). Among 103 microinvasive carcinoma samples analyzed by Kim et al. using IHC, there was no HER2-positive case [65].

**Tumor stage and HER2 positivity prevalence.** A meaningful and reliable analysis of the relationship between tumor stage and HER2 positivity prevalence could not be carried out as most of them included a mixture of several stages in varied, often unknown proportions, not always providing the mapping of HER2 status to the tumor stage of the participants. The main findings of the review are summarized in Table 2.

## Discussion and conclusion

In this work, we analyzed a large number of studies and showed that the prevalence of HER2 positivity in CC heavily depended on whether the standardized ASCO/CAP guidelines-compliant methodology was used. Based on the subset of ASCO/CAP compliant studies, we estimated the pooled prevalence of HER2 overexpression at 5.7% (CI 95%: 1.5% - 11.7%) and HER2 amplification at 1.2% (CI 95%: 0.0% to 5.8%). According to our findings, HER2 positivity rates above 10% can hardly be expected in unselected patients. As high degrees of statistical heterogeneity were observed, relying on 95% confidence intervals instead of point estimates may be more appropriate when interpreting the results of pooled analyses.

In comparison with other tumor sites where HER2 is already an established therapeutic target, the pooled HER2 overexpression rate in CC looks much lower than in malignancies with

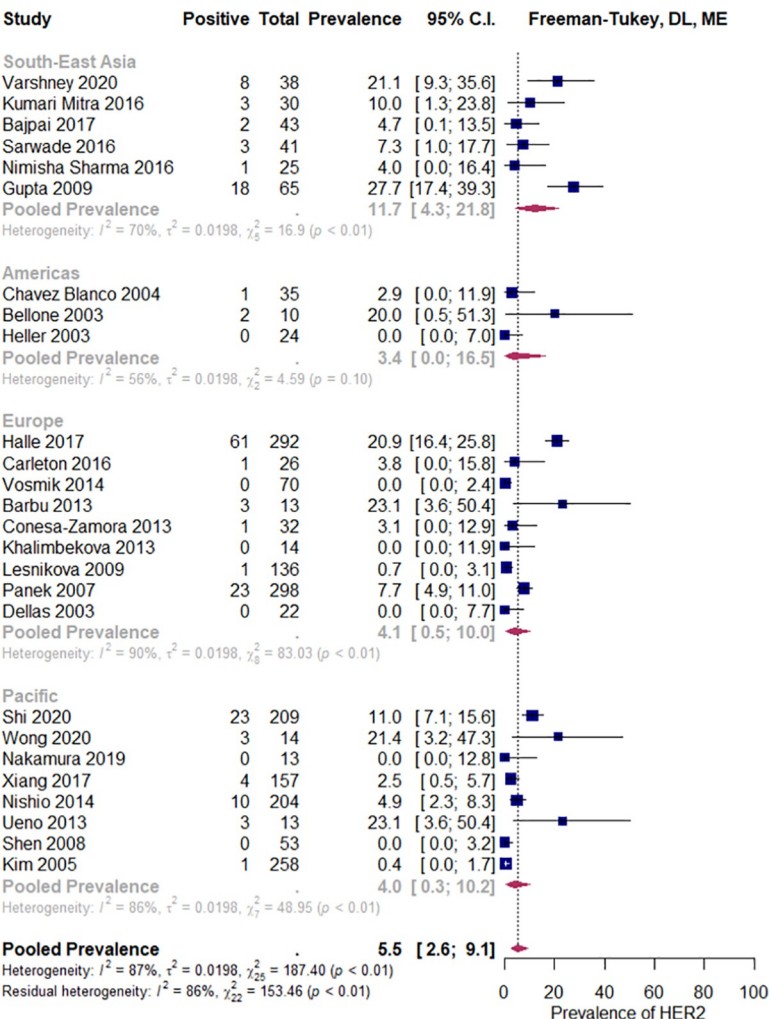

**Fig 6. Subgroup analysis by World Health Organization geographic region.** Abbreviations: DL = Der Simonian and Laird, ME = mixed effects.

the highest proportion of HER2-positive tumors like serous endometrial (47%), gastroesophageal (34%), and breast carcinoma (15% to 25%) [95, 96]. but slightly higher than in colorectal (2%) or lung cancer (3%) [97, 98]. This comparison may have limitations because tumors arising from different tissues have distinct patterns of HER2 amplification, and overexpression reflected in unequal Criteria for Positivity (≥50% in colorectal, ≥30% serous endometrial, ≥10% breast and gastric cancer) and different rates of heterogeneity [95].

The rate of HER2 amplification in our study looks quite low compared with data from online genomic databases. For example, in the curated datasets of non-redundant studies from the cBioPortal database, HER2 amplified tumors are 5.6% of all CC, 10.9% cervical adenocarcinomas, and 2.8% of squamous CC [99]. The cause of this discrepancy is unclear. In CC, the concordance between HER2 amplification by ISH and DNA-sequencing techniques seems to be insufficiently studied. The same may hold regarding the concordance between IHC and ISH in our study. Although the comparison is indirect, the pooled estimated HER2 positivity prevalence seems to be lower when determined by ISH than by IHC. As a possible explanation, Conesa-Zamora et al. suggested that the increased copy number of chromosomes 17 is due to polyploidy, frequently present in advanced stages and HPV associated tumors [26, 100].

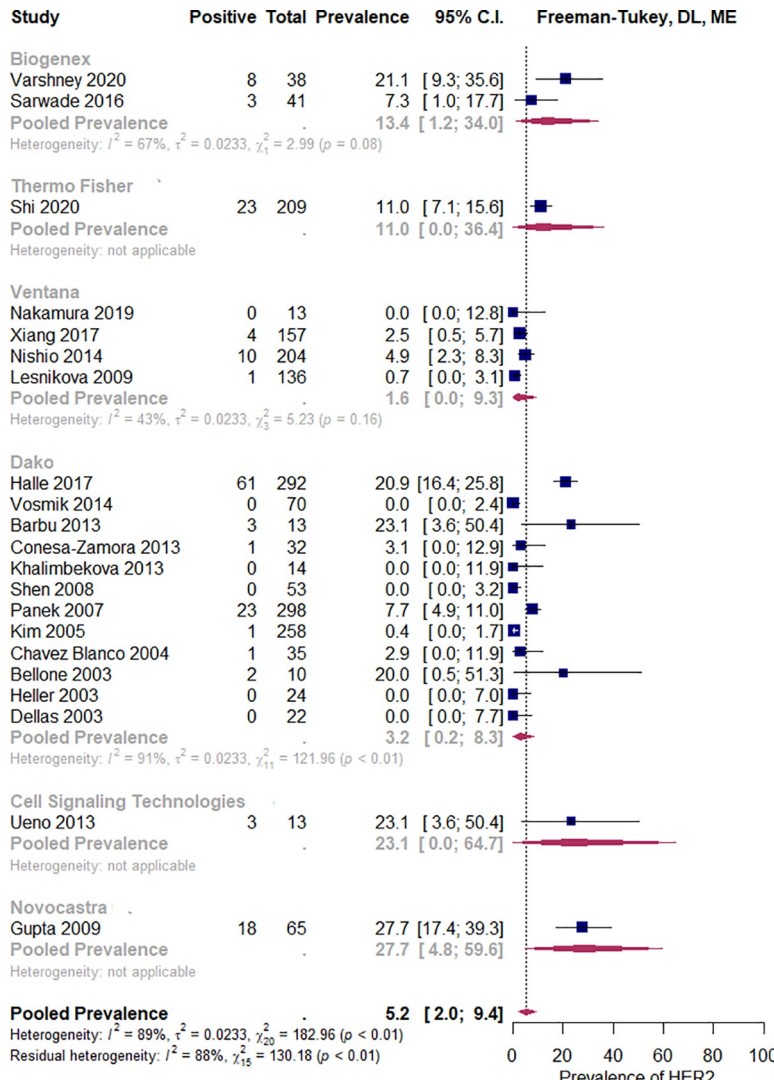

**Fig 7. Subgroup analysis by primary antibody brand.** Abbreviations: DL = Der Simonian and Laird, ME = mixed effects.

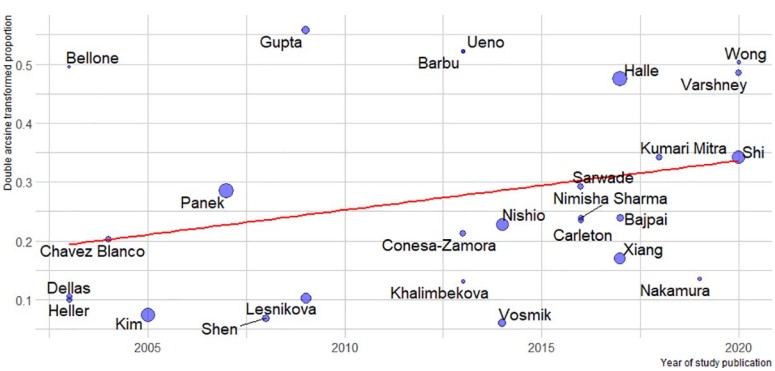

**Fig 8. Relationship between the year of study publication, size, and HER2 overexpression prevalence.**

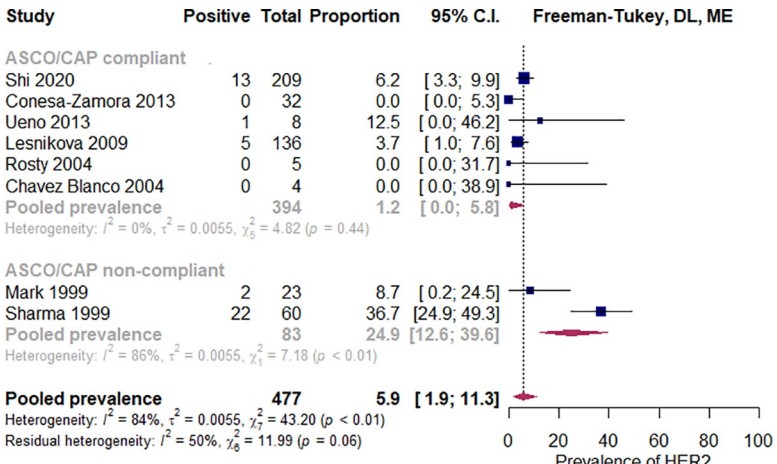

**Fig 9. Prevalence of HER2 amplification according to compliance with ASCO/CAP guidelines.** Abbreviations: ASCO = American Society of Clinical Oncology, CAP = College of American Pathologists, DL = Der Simonian and Laird, ME = mixed effects.

Unexpectedly, in our analysis, the trend to higher HER2 positivity rates in the subgroup of non-squamous histology compared to squamous CC has not reached statistical significance. The estimated pooled prevalence of HER2 positivity in squamous CC was above 4%. Although relatively low in general, it looks slightly higher than figures reported in squamous carcinomas of other primary sites [101, 102]. On the other hand, the non-squamous CC subgroup was composed of numerous, sometimes distinct entities (Fig 10 and S11 File). Although we failed to find (a) histologic subtype(s) with a particularly high HER2 positivity in this subgroup, it does not mean that such a subset could not be found in the future.

The impossibility to thoroughly explain the statistical heterogeneity is a significant limitation of our study. Many factors can potentially contribute to the observed heterogeneity. Disparities in HER2 expression in different histologic subtypes were discussed above [39]. An unequal racial/ethnic background could act as an effect modifier variable. Santin et al. found that women of African ancestry had a higher rate of HER2-positive serous endometrial carcinoma than women of other races [103]. In this regard, the numerical trend towards a greater prevalence of HER2 positivity in India observed in our study might deserve further investigation. Unknown biopsy-to-fixation and fixation-to-assay times, as well as sample handling during the pre-analytical stage, are limitations of the evidence included in the review. This study could not assess the impact of the tumor stage, which is another limitation of our work. In breast cancer, there is significant discordance in HER2 positivity between primary and metastatic sites. HER2 loss was observed in 21.3% and HER2 gain in 9.5% of cases. Discordance in HER2 status has also been documented between distinct metastatic sites [104]. If a similar phenomenon exists in CC, it could contribute to explaining the statistical heterogeneity in our study.

Intra-tumor HER2 heterogeneity may also affect the results of HER2 testing. Both the cluster and disperse types of heterogeneity have been described in breast cancer [21]. In serous endometrial carcinoma, heterogeneous HER2 protein expression defined as the presence of at least 2 degrees of difference in staining intensity in at least 5% cells was found in 50% of the cases classified as positive [95, 105]. Intra-tumor HER2 heterogeneity was also reported in gastroesophageal and, to a lesser extent, breast cancer [95, 106]. Other possible sources of statistical heterogeneity not addressed in this study are patients' age and treatments before the HER2 status determination. The relationship of the HER2 prevalence with the latter could not be investigated due to the lack

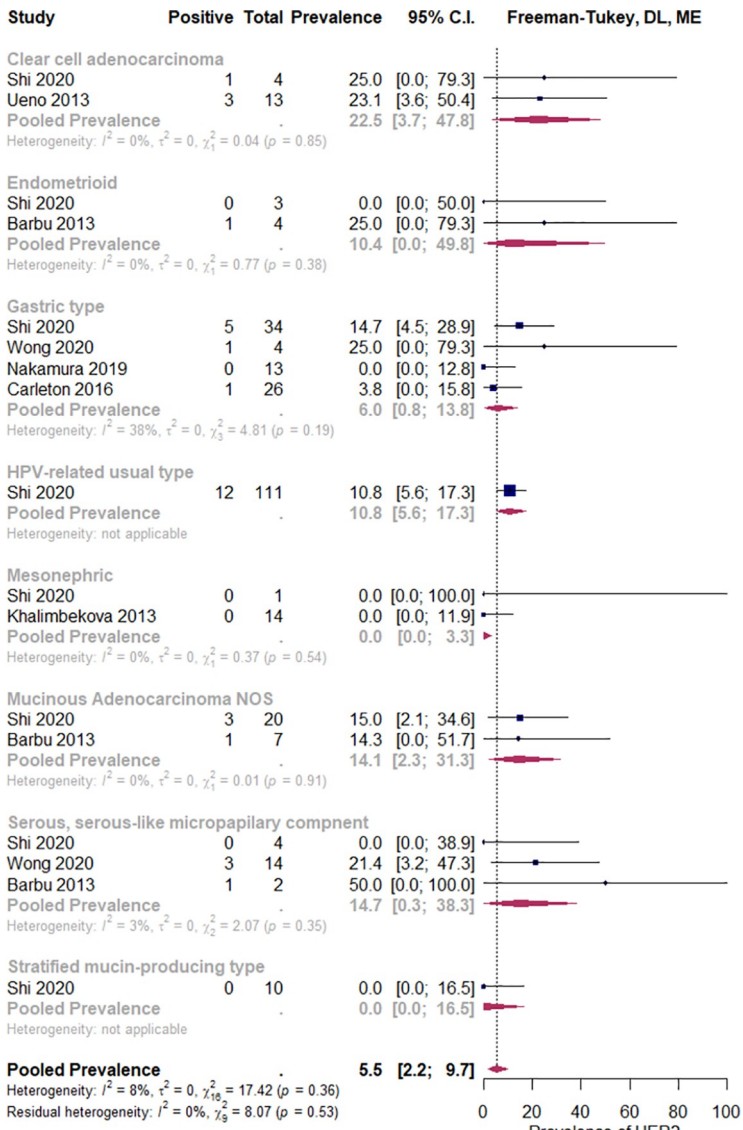

**Fig 10. HER2 overexpression prevalence in selected histologic subtypes.** Abbreviations: DL = Der Simonian and
Laird, ME = mixed effects.

of data in most studies. Our study's additional limitations are incomplete retrieval of the identified
research and pooling of not entirely homogeneous studies, concerning their populations and risk
of bias, despite the significant effort to override these issues.

A better standardization of the IHC procedure in CC may be desirable. Future research
should look into the concordance of HER2 positivity as determined by IHC at various cut-off
points, ISH, and next-generation sequencing, as well as how accurate each of these methods is
in predicting clinical benefit from HER2 targeted drugs. Of note, the ASCO/CAP guidelines
for breast cancer were developed largely based on the benefit of trastuzumab. If new drugs
active in patients with HER2-low-expressing tumors become available, the proportion of CC
patients with this potentially relevant actionable target may increase [107]. The accessibility of
some HER2 testing methods in countries where CC is prevalent may be an issue.

**Table 2. Summary of the main findings of the review.**

| | Estimated HER2 Pooled Prevalence (95% CI) | № of participants (studies) | P-value | Risk of bias |
|---|---|---|---|---|
| **All studies** | | | | |
| **IHC all studies** | 17.0% (11.7–23.0) | 5076 (62) | - | - |
| **ISH all studies** | 5.9% (1.9–11.3) | 477 (8) | | - |
| **By ASCO/CAP guidelines compliance** | | | | |
| **IHC ASCO/CAP compliant** | 5.7% (1.5–11.7) | 2135 (26) | < 0.001 | Low |
| **IHC ASCO/CAP non-compliant** | 27.0% (19.9–34.8) | 2941 (36) | | High |
| **ISH ASCO/CAP compliant** | 1.2 (0.0–5.8) | 394 (6) | 0.004 | Low |
| **ISH ASCO/CAP non-compliant** | 24.9% (12.6–39.6) | 83 (2) | | High |
| **By histologic subtype** | | | 0.054 | |
| **Squamous** | 4.1% (0.6–9.8) | 12 (1018) | | Low |
| **Non-squamous** | 10.3% (3.6–19.2) | 15 (467) | | Low |
| **By WHO geographic region** | | | 0.40 | |
| **Southeast Asia** | 11.7 (4.3–21.8) | 6 (242) | | Low |
| **Americas** | 3.4 (0.0–16.5) | 4 (95) | | Low |
| **Europe** | 4.1 (0.5; 10.0) | 8 (877) | | Low |
| **Pacific** | 4.0 (0.3; 10.2) | 8 (921) | | Low |

Abbreviations: ASCO = American Society of Clinical Oncology, CAP = College of American Pathologists, WHO = World Health Organization

*According to the compliance with the ASCO/CAP guidelines, HPV = Human Papilloma Virus.

In summary, our results suggest that the prevalence of HER2-positive tumors in CC is low but may still be meaningful, and variations in previously reported HER2 positivity rates are mainly related to methodological issues. To our knowledge, this is the first meta-analytic systematic review on the subject published so far. Our findings reduce uncertainty regarding the expected frequency of HER2-positive CC and help to better understand the biology of this tumor, as well as to guide decisions about the appropriateness of anti-HER2 drug studies for CC and assist in their design.

## Supporting information

**S1 File. Bib search.**
(DOCX)

**S2 File. Data items.**
(DOCX)

**S3 File.** A Funnel ihc all. B Funnel compliant. C Funnel ish.
(TIFF)

**S4 File. Study characteristics.**
(DOCX)

**S5 File. Geographic information.**
(DOCX)

**S6 File. Characteristics ish methods.**
(DOCX)

**S7 File. Clinical features.**
(DOCX)

**S8 File. Risk of bias.**
(XLSX)

**S9 File. All ihc.**
(TIFF)

**S10 File. Influence diagnostic.**
(HTML)

**S11 File.**
(DOCX)

**S12 File. PRISMA checklist.**
(DOCX)

**S13 File. R code.**
(DOCX)

## Acknowledgments

We would like to thank our families for their patience, the Centro Cochrane IECS, Cochrane Argentina, for providing free access to Covidence software, and Mr. Daniel Comandé, IECS librarian, for his help with the bibliographic search.

## Author Contributions

**Conceptualization:** Boris Itkin, Samanta Straminsky, Eduardo Daniel Adelchanow, Matias Pereyra, Gabriela Acosta Haab, Ariel Bardach.

**Data curation:** Boris Itkin, Agustin Garcia, Eduardo Daniel Adelchanow.

**Formal analysis:** Boris Itkin, Samanta Straminsky.

**Investigation:** Boris Itkin, Agustin Garcia, Samanta Straminsky, Eduardo Daniel Adelchanow, Matias Pereyra, Gabriela Acosta Haab, Ariel Bardach.

**Methodology:** Boris Itkin, Samanta Straminsky, Eduardo Daniel Adelchanow, Matias Pereyra, Gabriela Acosta Haab, Ariel Bardach.

**Project administration:** Boris Itkin, Ariel Bardach.

**Software:** Boris Itkin, Samanta Straminsky.

**Supervision:** Agustin Garcia, Samanta Straminsky, Ariel Bardach.

**Validation:** Boris Itkin, Samanta Straminsky, Eduardo Daniel Adelchanow, Matias Pereyra, Gabriela Acosta Haab, Ariel Bardach.

**Visualization:** Boris Itkin.

**Writing – original draft:** Boris Itkin, Agustin Garcia, Ariel Bardach.

**Writing – review & editing:** Boris Itkin, Agustin Garcia, Samanta Straminsky, Eduardo Daniel Adelchanow, Matias Pereyra, Gabriela Acosta Haab, Ariel Bardach.

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
