## [Decision Letter · Decision Letter 0]

5 Aug 2021

PONE-D-21-12748

Prevalence of HER2 overexpression and amplification in uterine cervical cancer: a systematic review and a meta-analysis

PLOS ONE

Dear Dr. Itkin,

Thank you for submitting your manuscript to PLOS ONE. After careful consideration, we feel that it has merit but needs some minor revisions. Therefore, we invite you to submit a revised version of the manuscript that addresses the points raised during the review process.

We look forward to receiving your revised manuscript.

Kind regards,

Mona Pathak, PhD

Academic Editor

PLOS ONE

Journal Requirements:

2. Please ensure that you refer to Figure 8 in your text as, if accepted, production will need this reference to link the reader to the figure

Additional Editor Comments (if provided):

I appreciate the authors for their nice efforts in planning, conducting, and presenting this systematic review and meta-analysis.

Reviewers' comments:

Reviewer's Responses to Questions

**Comments to the Author**

1. Is the manuscript technically sound, and do the data support the conclusions?

Reviewer #1: Yes

Reviewer #2: Yes

2. Has the statistical analysis been performed appropriately and rigorously? 

Reviewer #1: Yes

Reviewer #2: Yes

3. Have the authors made all data underlying the findings in their manuscript fully available?

Reviewer #1: Yes

Reviewer #2: Yes

4. Is the manuscript presented in an intelligible fashion and written in standard English?

Reviewer #1: Yes

Reviewer #2: Yes

5. Review Comments to the Author

Reviewer #1: Article review (Prevalence of HER2 overexpression and amplification in uterine cervical cancer: a systematic review and a meta-analysis, PONE-D-21-12748):

Summary

This is a systematic review and meta-analysis on the prevalence of overexpression or amplification of HER2 in cervical cancer. Overall, this is a well-designed study that provides essential information that could assist in designing clinical trials on targeting HER2 in cervical cancer. This includes an estimate of the prevalence of HER2 alterations and the impact of ASCO/CAP compliant methodology on the HER2 positivity rate.

Comments/Revisions

1. The study is methodologically sound with high-quality data and statistical analysis. There are few grammar and phrasing errors, and professional editing is recommended

2.The term uterine cervical cancer should be replaced by cervical cancer in the title and throughout the manuscript.

3. HER2 targeting has been proven to be an effective therapeutic approach in breast and uterine serous carcinoma. The clinical data in cervical cancer is limited. SUMMIT is a phase II basket trial that included cervical cancer patients and demonstrated evidence of activity. In this preliminary phase of drug development, comparisons to other disease sites that HER2 has an established role are needed. The authors mention heterogeneity in breast and uterine serous cancer, but they should present more specific data on the HER2 expression in breast and uterine serous carcinoma. More information in the ‘Discussion’ is needed to discuss the methodology used and prevalence of overexpression and amplification in these disease sites.

Recommendation

Minor revision

Reviewer #2: Dear Authors:

Thank you for providing so interesting, consolidated data regarding HER2 positivity in such a relevant disease worldwide. I would also like to highlight your cautious and high-quality approach for summarizing observational data by means of systematic review of the literature, for which I have no comments.

I have the following minor comments. I hope they might help for the purpose of your manuscript.

1. Given that PLOSOne public might not be expert on the oncology field, I would consider further contextualization of Cervical Cancer issues in some additional detail. For example:

-What is the disease burden worldwide that makes this study topic relevant?

-What implications would HER2 overexpression detection have either via IHC or FISH? What about diagnostic precision and accessibility in CC?

2. Human Papillomavirus has been clearly related as etiologic risk factor for developing Cervical Cancer. Usual approaches towards controlling or eliminating infection have shown efficacy for preventing CC, but this might have hindered research on other disease mechanisms in the past. Your approach is innovative as it highlights a hypothetical potential prognostic and/or predictive factor in CC, which may contribute as part of multimodal cancer treatment. It would be very interesting if the authors could comment on this relationship so that potential new research could address current knowledge gaps based on your findings (See Conesa-Zamora P et al.Exp Mol Pathol. 2013 Oct;95(2):151-5.)

6. PLOS authors have the option to publish the peer review history of their article (what does this mean?). If published, this will include your full peer review and any attached files.

Reviewer #1: No

Reviewer #2: **Yes: **Andres Mauricio Acevedo

---

## [Author Response · Author response to Decision Letter 0]

11 Sep 2021

Dear Reviewers

Thank your for your valuable comments.

We provided answers to your questions and modified the manuscript acording your suggestions.

Very best regrads

The authors

---

## [Editor Report · Decision Letter 1]

15 Sep 2021

Prevalence of HER2 overexpression and amplification in uterine cervical cancer: a systematic review and a meta-analysis

PONE-D-21-12748R1

Dear Dr. Itkin,

We’re pleased to inform you that your manuscript has been judged scientifically suitable for publication and will be formally accepted for publication once it meets all outstanding technical requirements.

Kind regards,

Mona Pathak, PhD

Academic Editor

PLOS ONE

---

## [Editor Report · Acceptance letter]

22 Sep 2021

PONE-D-21-12748R1 

Prevalence of HER2 overexpression and amplification in cervical cancer: a systematic review and meta-analysis 

Dear Dr. Itkin:

I'm pleased to inform you that your manuscript has been deemed suitable for publication in PLOS ONE. Congratulations! Your manuscript is now with our production department. 

Kind regards, 

on behalf of

Dr. Mona Pathak 

Academic Editor

PLOS ONE